# Molecular Mapping of Urinary Complement Peptides in Kidney Diseases

**DOI:** 10.3390/proteomes9040049

**Published:** 2021-12-13

**Authors:** Ralph Wendt, Justyna Siwy, Tianlin He, Agnieszka Latosinska, Thorsten Wiech, Peter F. Zipfel, Aggeliki Tserga, Antonia Vlahou, Harald Rupprecht, Lorenzo Catanese, Harald Mischak, Joachim Beige

**Affiliations:** 1Department of Nephrology and Kuratorium for Dialysis and Transplantation (KfH) Renal Unit, Hospital St. Georg, 04129 Leipzig, Germany; Ralph.Wendt@sanktgeorg.de; 2Mosaiques Diagnostics GmbH, 30659 Hannover, Germany; siwy@mosaiques-diagnostics.com (J.S.); he@mosaiques-diagnostics.com (T.H.); latosinska@mosaiques-diagnostics.com (A.L.); mischak@mosaiques-diagnostics.com (H.M.); 3Nephropathology Section, Institute of Pathology, University Medical Center, 20246 Hamburg, Germany; t.wiech@uke.de; 4Institute of Microbiology, Friedrich-Schiller-University, 07743 Jena, Germany; peter.zipfel@hki-jena.de; 5Department of Infection Biology, Leibniz Institute for Natural Product Researach and Infection Biology, 07745 Jena, Germany; 6Biomedical Research Foundation, Academy of Athens, Department of Biotechnology, 11527 Athens, Greece; tserga@bioacademy.gr (A.T.); vlahoua@bioacademy.gr (A.V.); 7Department of Nephrology, Klinikum Bayreuth GmbH, 95447 Bayreuth, Germany; harald.rupprecht@klinikum-bayreuth.de (H.R.); lorenzoriccardo.catanese@gmail.com (L.C.); 8Department of Internal Medicine II, Martin-Luther-University Halle-Wittenberg, 06108 Halle (Saale), Germany

**Keywords:** complement, peptide, urine, biomarker, kidney disease, proteomics, CE-MS, capillary electrophoresis

## Abstract

Defective complement activation has been associated with various types of kidney disease. This led to the hypothesis that specific urine complement fragments may be associated with kidney disease etiologies, and disease progression may be reflected by changes in these complement fragments. We investigated the occurrence of complement fragments in urine, their association with kidney function and disease etiology in 16,027 subjects, using mass spectrometry based peptidomics data from the Human Urinary Proteome/Peptidome Database. Twenty-three different urinary peptides originating from complement proteins C3, C4 and factor B (CFB) could be identified. Most C3-derived peptides showed inverse association with estimated glomerular filtration rate (eGFR), while the majority of peptides derived from CFB demonstrated positive association with eGFR. Several peptides derived from the complement proteins C3, C4 and CFB were found significantly associated with specific kidney disease etiologies. These peptides may depict disease-specific complement activation and could serve as non-invasive biomarkers to support development of complement interventions through assessing complement activity for patients’ stratification and monitoring of drug impact. Further investigation of these complement peptides may provide additional insight into disease pathophysiology and could possibly guide therapeutic decisions, especially when targeting complement factors.

## 1. Introduction

The urinary proteome holds information on disease and disease pathophysiology [1]. Significant association of specific urinary peptides was demonstrated in large studies that frequently included >1000 subjects in several disease areas, among others, in kidney disease [2,3], cardiovascular disease [4,5] and several types of cancer [6,7,8], typically applying capillary electrophoresis coupled mass spectrometry (CE-MS) [9,10].

The complement system is a crucial effector of the innate immune system and involved in mechanisms of the adaptive immune system. It is activated via 3 different pathways, comprised of over 30 different plasma or membrane-bound proteins [11,12]. The complement system is increasingly recognized as an important mediator in the context of kidney diseases [13]. Immune complexes are strong activators of the classical complement pathway and are involved in various forms of glomerulonephritis, e.g., lupus nephritis (LN). Dysregulation of the alternative pathway of complement activation may lead to atypical hemolytic uremic syndrome (aHUS) and C3-glomerulopathy (C3G) [14]. All complement activation pathways converge at the level of C3-convertase. Complement activation as well as modulation of complement activity is exerted on the tissue or cell surfaces, or in the circulation. Most studies of complement association with disease involve biomarkers of complement functions or complement factors in serum or plasma of their deposition on the tissue surface. Studies on complement component excretion in urine are scarce and might add important information on complement status and function in different diseases.

Due to complement being centrally involved in multiple chronic kidney disease (CKD) etiologies, multiple drug candidates interfering with complement activity have recently been developed and are currently undergoing testing. More than 25 clinical trials in this context are currently listed (www.clinicaltrials.gov) (accessed on 15 July 2021), further described in recent review articles [15,16]. Targeting complement C5 (e.g., Crovalimab, Ravulizumab, Eculizumab, Cemdisiran) and complement factor B (CFB) (e.g., LNP023, IONIS-FB-LRx), are most frequently investigated. Drugs targeting C5 are investigated among others in the context of aHUS, IgA nephropathy (IgAN), LN and acute kidney injury due to COVID-19; while those targeting CFB are studied, for example, in C3G, IgAN and membranous glomerulonephritis (MGN). Other drugs being currently tested target C1r and C1s (C1-INH for kidney transplantation), C3 (APL-2 for IgAN, LN, MGN, C3G, membranoproliferative glomerulonephritis (MPGN)), C5a (IFX-1 for anti-neutrophil cytoplasmic antibody (ANCA)-associated vasculitis), C5aR1 (Avacopan for C3G and ANCA- associated vasculitis), complement factor D (ACH-4471 for C3G, MPGN and ALXN2050 for renal impairment) and mannan-binding lectin serine protease 2 (OMS721 for IgAN, LN, MGN, C3G). Although multiple trials are ongoing, development of complement therapeutics remains challenging, mainly due to differences in complement pathway activities between individuals and complex pathogenic mechanisms [15]. To address these issues, development of personalized complement interventions is being advocated, with a focus on future monitoring of response to therapy. The availability of a comprehensive database of urine peptide and low molecular weight protein data from currently >80,000 subjects [17] has generated the opportunity to perform in-depth analysis of urine peptides and low molecular weight proteins, based on large numbers of patient-derived datasets. Investigations have been performed either in the context of specific diseases or physiology (e.g., heart failure or obesity [4,18]), or specific peptides (e.g., polymeric immunoglobulin receptor or thymosin beta-4 [19,20]). Given the association of complement fragments with kidney disease, the hypothesis was generated that complement-derived peptides in urine may allow monitoring of complement activation. Such monitoring of dynamic changes in complement system, rather than measuring absolute levels of proteins in clinical samples, may better reflect the disease activity and pathogenic mechanisms, thus guiding intervention. Therefore, we investigated the urinary peptidome for the presence of complement-derived fragments and in a next step assessed the distribution of these peptides in different CKD etiologies, to identify potential specific association of complement fragments with CKD etiologies.

## 2. Materials and Methods

### 2.1. Proteomic Investigation, Data Retrieval and Diagnosis Assessment

Urinary proteome data obtained by CE-MS were assessed in the Human Urine Proteome/Peptidome Database [10]. The CE-MS technology applied has been described in detail including reproducibility, repeatability, procedures for sample preparation, data evaluation and normalization [21]. In short, CE-MS analysis was performed with a P/ACE MDQ CE (Beckman Coulter, Brea, CA, USA) coupled to a micro-TOF-MS (Bruker Daltonic, Bremen, Germany). RAW MS data were evaluated using MosaFinder applying a probabilistic clustering algorithm and using both isotopic distributions and conjugated masses for charge state determination [17]. Normalization of the peptide signal intensity was based on 29 collagen fragments that are generally not affected by disease and serve as internal standards, described in detail in [22]. This normalized intensity is referred in this manuscript as the relative peptide abundance. This database currently contains urinary proteome/peptidome information on >80,000 samples, assessing currently 4080 sequenced peptides and low molecular weight proteins [17], and anonymized clinical information of participants enrolled in the studies.

MS/MS experiments were conducted using an Ultimate 3000 nano-flow system (Dionex/LC Packings, Camberley, UK) connected to an LTQ Orbitrap hybrid MS (Thermo Fisher Scientific, Waltham, MA, USA) equipped with a nano-electrospray ion source [23]. The MS was operated in data-dependent mode to automatically switch between MS and MS/MS acquisition. RAW MS data were searched against the Swiss-Prot human database using Proteome Discoverer 2.4 and the SEQUEST search engine. Relevant settings were: no fixed modifications, oxidation of methionine and proline as variable modifications. The minimum precursor mass was set to 790 Da, maximum precursor mass to 6000 Da with a minimum peak count of 10. The high-confidence peptides were defined by cross-correlation (Xcorr) score > 1.9. Precursor mass tolerance was 5 ppm and fragment mass tolerance was 0.05 Da. For further validation of obtained peptide identifications, the correlation between peptide charge at the working pH of 2 and CE-migration time was utilized to minimize false-positive identification rates. Raw data from the representative LC-MS/MS runs used for sequencing of complement peptides (as presented in Appendix A) are available at Zenodo (doi:10.5281/zenodo.5713591).

Underlying diagnosis related to histopathology findings were obtained by kidney biopsy. If biopsy was not indicated as for instance in the majority of cases with acute kidney failure and kidney stones, diagnoses were obtained by assessment of the available clinical information. All proteomics datasets from healthy controls or subjects with kidney disease with the information about disease condition, where at least one complement peptide could be detected, were extracted. Based on these criteria, 16,027 anonymized datasets were included in the analysis.

### 2.2. Assessment of eGFR and Correction for Proteinuria

Estimated glomerular filtration rate (eGFR) was calculated from serum creatinine using the Chronic Kidney Disease Epidemiology Collaboration (CKD-EPI) equation [24]. Complement peptides in urine are in part the result of glomerular filtration, and in part the result of proteinuria, and both of these processes contribute to the final abundance of urinary peptides. Thus, to correct for proteinuria (g/g creatinine) (specifically the fraction of peptides being a result of the proteinuria), impact of proteinuria on the complement abundance was estimated, and then subtracted from the peptide abundance. In brief, first, the relative abundance of all peptides originating from the same protein was summed up (“combined abundance”). This approach is further justified since most peptides observed are from the same region of the respective complement protein, and hence, are mutually exclusive (only one of these specific peptides can be generated per one complement molecule). The data were log transformed and correlation of the combined abundance with log proteinuria was assessed. Specifically, from the regression equation ((combined abundance) = a + b × (proteinuria)), the value of “b”, representing the impact of proteinuria on the abundance of complement peptides in urine, was first determined. This analysis was performed separately for each complement protein. Subsequently, the respective value of “b” was applied for each respective peptide as follows: (log peptide relative abundance corrected) = (log peptide relative abundance observed) – b × (log proteinuria).

### 2.3. Statistics

The association of proteomic data for all detectable complement fragments together with clinical data were investigated using SPSS 22.0 software (IBM). Correlation analysis was performed based on log transformed values using the Spearman’s rank correlation.

## 3. Results

A total of 23 complement fragments could be identified with high confidence in human urine. Tandem mass spectra from all 23 peptides are shown in Appendix A. These urinary complement peptide fragments are presented in Table 1.

CE-MS datasets with available information on disease and at least one of the complement peptides detectable were extracted from the database. This resulted in 16,027 anonymized datasets that could be retrieved and employed in further analysis. A list of the number of subjects per disease etiology/condition is presented in Table 2.

First, potential correlation of complement fragments with eGFR was investigated. Data on eGFR were available from 8388 datasets. The results are presented in Table 1. Most complement C3 peptides demonstrate a weak, yet significant inverse association with eGFR. Correlation of the CFB fragments with eGFR was inconsistent: both inverse and direct association were observed. As examples, the association of the most abundant peptide in urine from complement C3, L_982_QGTPVAQMTEDAVDAERLKHL_1003_ (e12939), and from CFB, L_235_SSLTETIEGVDAEDGHGPGEQ_257_ (e11594) with eGFR is shown in Figure 1A,C. Most of the complement C4A and B peptides did not show association with eGFR.

The urinary excretion of all detected complement fragments stratified by disease etiology/condition is shown in Figure 2. Due to the complexity of the data matrix depicting the abundance of 23 peptides in 22 diseases/conditions, not all data points can be visualized in the Figure 2. To enable detailed inspection, all underlying data are also presented in Appendix A. Shown are individual peptide excretion levels normalized to healthy controls. This was calculated by division of mean abundances per disease/condition with the mean corresponding abundance in healthy controls. Highest relative levels (in reference to healthy controls) for LQGTPVAQMTEDAVDAERLKHL (e12939, complement C3) excretion were seen in minimal change disease (MCD), followed by nephrotic syndrome (NS), focal segmental glomerulosclerosis (FSGS), MGN, LN, diabetic kidney disease (DKD), IgAN, MPGN and C3G.

Due to the high levels of excretion of complement C3-derived peptides in IgAN and the availability of a substantial number of follow-up data, we investigated the corresponding representative peptide in the independent PersTIgAN cohort [25]. In this cohort of 209 patients, LQGTPVAQMTEDAVDAERLKHL (e12939, complement C3) was significantly increased in the tertile of patients with the highest eGFR loss during follow-up, in comparison to the patients with the lowest eGFR loss (*p* = 0.0225).

The significant inverse correlation between eGFR and C3 peptides may not be a direct result of (a change in) the glomerular filtration, but due to different levels of proteinuria. Consequently, we investigated association of the abundance of complement peptides with proteinuria. Data on proteinuria were available from 4086 datasets. Examples on the association of peptide abundance and proteinuria for the above mentioned two peptides (e12939, e11594) are provided on Figure 1B,D. As shown, a significant and relevant (rho = 0.44) direct association of complement C3 peptide (e12939) with proteinuria could be detected. Association could in general also be detected when investigating the other complement C3 peptides (Table 1). As also evident from Figure 1, in general association with eGFR and proteinuria were found of opposite orientation, as expected, since eGFR is inversely associated with proteinuria.

These observed associations of proteinuria with the complement C3 fragments indicate that the abundance of complement fragments in urine is in part the result of glomerular filtration of the respective peptide, and in part the result of proteinuria.

The sequences of the observed complement peptides indicate that these originate in general from one specific region, with substantial overlap (see amino acid positions in Table 1). These peptides are mutually exclusive; only one of these peptides can be generated from a single complement molecule. Based on this observation, the relative abundance values of the peptides belonging to the respective complement protein were combined (their relative abundance was added) into a single variable.

The results of this approach, the distribution of the combined peptide abundances per complement component are shown in Figure 3A, limited to those datasets where proteinuria values were available. As expected, a strong correlation was observed for the combined complement C3 peptides with proteinuria. A weak yet significant correlation was also detected when investigating complement C4A and B, and a weak inverse correlation could be detected for CFB (shown in Appendix A). This “combined abundance” was used to correct for proteinuria, also to avoid overfitting when correcting based on each single peptide. The “combined abundance” in the different CKD etiologies corrected for proteinuria is shown in Figure 3B. When applying this correction for the respective combined abundance levels, correlation with proteinuria was lost (see Appendix A).

Correction for proteinuria calculated based on the change in “combined abundance” was applied for each peptide from each patient, and the distribution in the subjects where proteinuria was available was investigated in the different CKD etiology/condition, with and without correction for proteinuria. The results are presented in Figure 4. In the case of C3 fragments after adjustment for proteinuria, highest excretion was seen in MCD, C3G, DKD, FSGS, MGN, LN and vasculitis. For CFB fragments corrected for proteinuria, high excretion was seen e.g., in acute kidney injury and MPGN.

## 4. Discussion

This is the first study to present a comprehensive investigation of complement fragment excretion in urine in a large cohort of patients with different kidney disease etiologies and/or clinical scenarios. Complement peptide excretion may serve as a surrogate of complement system activation status.

Currently, limited data are available on the urinary excretion of complement components. The majority of complement factors are of molecular mass above the threshold for glomerular filtration in the range of 15–50 kDa [26]. The larger complement proteins are C1q (410 kDa), C4 (210 kDa), C4b (200 kDa), C3 (180 kDa) as well as C5 (180 kDa), composed of polypeptide chains usually between 22.5 and 40 kDa [27]. Complement is activated by proteolysis, resulting in cleavage products, some of which pass through the glomerular basement membrane. However, this has not been systematically investigated yet.

Low levels of complement C3 were detected in urine of healthy volunteers [28]. In this study, the authors suggested that 90% of complement C3 is reabsorbed at the proximal tubule. In the same study, significant urinary complement C3 excretion in MPGN, MGN, IgAN and LN was shown, similar to the findings in our investigation. Sakakibara and colleagues also observed a significant correlation between urinary excretion of complement C3 with intraglomerular immune complex deposition, especially along the glomerular capillary wall, less when the deposits were solely at the mesangium.

In our study, proteinuria was significantly and directly correlated with urinary C3 derived peptide abundance, while this was not observed for CFB. This ambiguous regulation of complement fragments with respect to proteinuria and eGFR may be owed to the role of the different components in complement activation: while CFB is involved in the activation via the alternative pathway, C3 is a key component in the process of amplification. As such, an association of CFB activation with specific disease aetiology seems plausible, but probably not with disease severity (proteinuria). In contrast, amplification is likely associated with disease severity and progression, which may be an explanation for the observed association of urinary C3 peptides with proteinuria. An alternative explanation may be that the plasma level of complement C3 is about 1.0–1.5 g/L, while CFB is present in much lower amount (ca. 200 mg/L). Thus, proteinuria will lead to substantial amount of complement C3 in urine, while a ~5-fold lower amount of urinary CFB as a result of proteinuria is expected. The minor yet significant inverse correlation of CFB peptide abundance with proteinuria may be due to a small, yet detectable signal suppression caused by the unspecific proteinuria.

When comparing urinary with plasma complement levels in different diseases in a relatively small study based on immunological detection, highest amount of urinary complement (iC3b, Bb) were reported in FSGS and DKD, with almost undetectable levels in healthy individuals [29], in agreement with our findings.

For most urine complement C3 fragment levels, we observed an inverse association with eGFR. This inverse association was generally not observed for complement C4, while CFB fragments generally showed a weak positive correlation with eGFR (Table 1). Lower eGFR is associated with increased proteinuria in the dataset investigated (see Appendix A). It seems plausible that the observed association of specific C3 complement fragments with eGFR is in part the result of proteinuria. However, even upon correction for proteinuria, a pronounced increase of complement C3 peptides is observed associated with distinct CKD etiologies, especially with MCD and FSGS.

When investigating the excretion of the most abundant peptide of C3 (LQGTPVAQMTEDAVDAERLKHL) in different diseases after normalization of individual excretion values to the median abundance of healthy controls (Figure 2), the highest relative levels were documented in MCD, FSGS, IgAN and LN, entities with known involvement of complement activation in the pathophysiology of the diseases. As evident especially in Figure 4, high levels of specific complement C3 peptides are detectable in MGN. This intense excretion of specific complement peptides in MGN may seem surprising and for a long time the disease has not been associated with increased complement activity. Nevertheless, evidence by now suggests that the complement system plays an important role in MGN [30], even though the complement pathways in MGN remain unexplored to some extent. Proteome analyses of kidney tissue samples from patients with MGN also showed significant complement activation and evidence of complement peptides from C3 and C4-related pathways [31]. Urinary excretion of complement activation products (sC5b-9, C3d) has been shown to correlate with disease activity [32,33,34].

Complement peptides detected in urine do not cover the entire complement molecules, as would be expected if generation and secretion into urine was by chance and random. Instead, specific regions of the respective complement proteins are covered, presented in most cases by multiple peptides. This is most pronounced for complement factor B, where all peptides detected are from the same region, amino acid positions 234–257. Upon closer examination of the regions represented from all three complement proteins, it appears that these are in each case regions that do not have an active role in the complement activation, but rather are removed upon activation. In fact, all CFB peptides originate from the same region and represent Bb fragments, being a part of the C3 convertase and constituting a stable complex with C3b on target surfaces. As such, it is an attractive hypothesis that these peptides represent the final products of complement activation, may therefore display previous complement activation, and could be exploited as non-invasive biomarkers for such activation.

We found increased levels of most urinary complement peptides in LN in comparison to systemic lupus erythematosus (SLE), also after adjustment for proteinuria (Figure 3 and Figure 4). Analysis of urinary complement excretion may be of interest in LN patients since urinary C3d was superior to plasma C3, C4d, Bb, C5b-9 and anti-double-stranded DNA antibody in distinguishing patients with LN from those without acute LN [35].

The complement system is increasingly appreciated as mediator and contributor to renal inflammation and tissue damage in IgAN, but the mechanism of activation and the contributing role are not well defined and in part unclear [36,37]. In recent years, a role of the alternate and lectin pathway of complement activation have been proposed in the pathogenesis of IgAN. In our analysis, we detected high levels of C3 and CFB fragments in urine (Figure 2), factors that can be attributed to the alternate pathway of complement activation. In the PersTIgAN cohort [25], urinary C3 excretion was significantly higher in patients exhibiting higher eGFR loss vs. those with lower eGFR loss during follow-up.

In vasculitis urinary excretion of complement fragments was increased compared to healthy controls. The majority of patients in this group had renal involvement of ANCA-associated vasculitis. Despite the microscopic “pauci-immune” nature of the glomerulonephritis in immunofluorescence evaluation, the available evidence indicates that activation of the complement system through the alternative pathway is necessary for the development of ANCA-associated vasculitis and is also associated with prognosis of the disease [38].

In patients with DKD, complement fragment excretion was higher than in healthy controls, but also higher than in diabetic patients with preserved renal function (Figure 2, Figure 3 and Figure 4). Especially, excretion of C3 complement fragments was higher in patients with DKD than in diabetic patients without kidney involvement. The role of complement activation in DKD and its importance for renal pathology and the progression of kidney damage is becoming increasingly evident [39,40]. Transcriptome and immunohistochemical analyses of human and mouse kidney biopsies revealed increased glomerular and/or tubular deposition of C3 and CFB compared to healthy controls [41,42,43,44]. C4A mRNA was reported increased in glomeruli of DKD patients compared to healthy controls [44]. These data reinforce the hypothesis that complement activation contributes to pathogenesis of kidney diseases and potentially induces tissue damage through inflammation.

Data from mouse studies demonstrate the importance of complement in kidney diseases. Glomerular C3 deposits have been detected increased in type 1 diabetes mellitus non-obese diabetic mouse [45] and in OVE26 diabetic mouse [46]. C3 and C3 cleaved products were detected in the glomeruli of adriamycin-treated mice mimicking human FSGS [47]. C4 was associated with LN mouse models [48]. As such, the human data from this study could serve as a basis to investigate complement and its fragments in animal models in more detail.

Our study has several limitations. First, we could not control for confounders that might influence complement activation and complement peptide excretion (e.g., infections, different types of LN, heterogeneous variants of FSGS, therapy) in the different subcategories of diseases. Second, we did not have values for proteinuria and eGFR available for all patients in the study. Nevertheless, the very large number of datasets with associated proteinuria and/or eGFR certainly represents a major strength of the report. Third, the change in urinary output of a complement fragment does not automatically associate it with a contributing role in the disease processes.

## 5. Conclusions

This is the first study comprehensively assessing urinary complement peptides in a large cohort of healthy controls and patients with different kidney diseases. We provide evidence for the activation–excretion of complement components in urine in several human kidney diseases. Multiple complement-derived peptides were significantly associated with kidney function, and with specific kidney disease aetiologies. Such peptides might provide useful information on disease status, and serve as a diagnostic tool, e.g., for discrimination between active and non-active disease. Based on the data available, and considering the need for developing personalised complement therapies, proteomic screening of these complement factors in urine could support patient selection and monitoring the drug impact in trials targeting complement components, and guide therapeutic decisions in a personalized manner.

## Figures and Tables

**Figure 1 proteomes-09-00049-f001:**
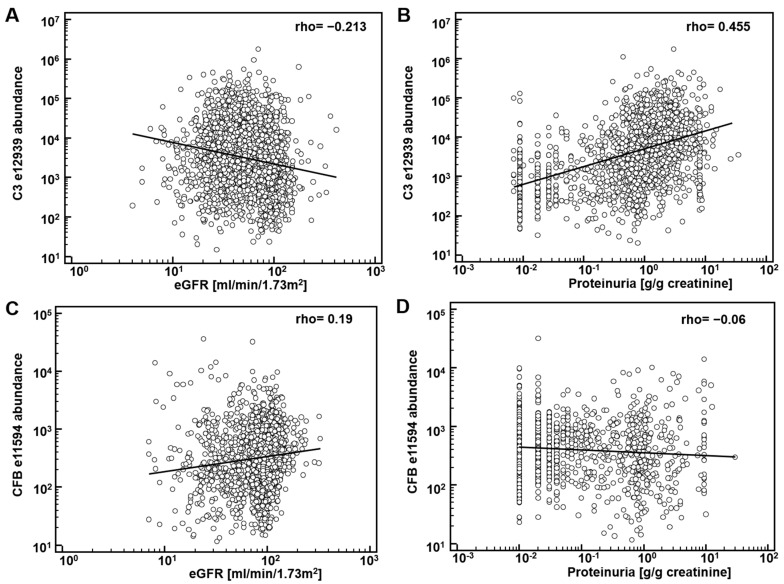
Association of complement-derived peptides with eGFR and proteinuria: (**A**) association of the most abundant peptide in urine from complement C3 (C3), L_982_QGTPVAQMTEDAVDAERLKHL_1003_ (e12939) with eGFR; (**B**) association of e12939 with proteinuria; (**C**) association of L_235_SSLTETIEGVDAEDGHGPGEQ_257_ (e11594), from Complement factor B (CFB) with eGFR; (**D**) association of e11594 with proteinuria.

**Figure 2 proteomes-09-00049-f002:**
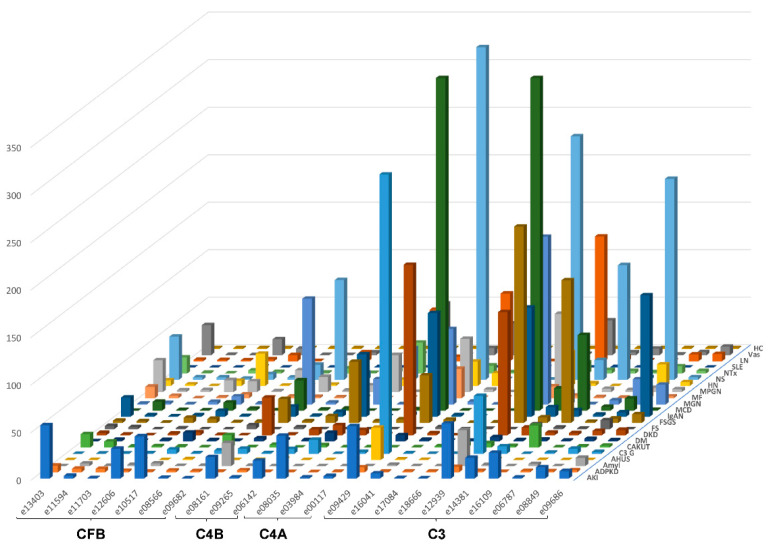
Average relative peptide abundances for each complement fragment per disease or condition, normalized by average relative abundance in healthy controls. The ID of the peptides is given as indicated in Table 1. Peptides are sorted according to their origin: first, peptides from complement factor B (CFB), then C4B and C4A, followed by C3. Within each protein, peptides are sorted based on the position of first amino acid. Abbreviations: ADPKD: Autosomal dominant polycystic kidney disease; AHUS: Atypical hemolytic uremic syndrome; AKI: Acute Kidney Injury; Amyl: Amyloidosis; C3G: C3 glomerulopathy; CAKUT: Congenital anomalies of the kidney and urinary tract; DKD: Diabetic kidney disease; DM: Diabetes Mellitus; FS: Fanconi syndrome; FSGS: focal segmental glomerulosclerosis; HC: Healthy control; HN: Hypertensive nephrosclerosis; IgAN: IgA nephropathy; LN: lupus nephritis; MCD: minimal change disease; MF: Morbus Fabry; MGN: membranous glomerulonephritis; MPGN: membranoproliferative glomerulonephritis; NS: Nephrotic syndrome; NTx: Kidney transplantation; SLE: Systemic lupus erythematosus; Vas: Vasculitis.

**Figure 3 proteomes-09-00049-f003:**
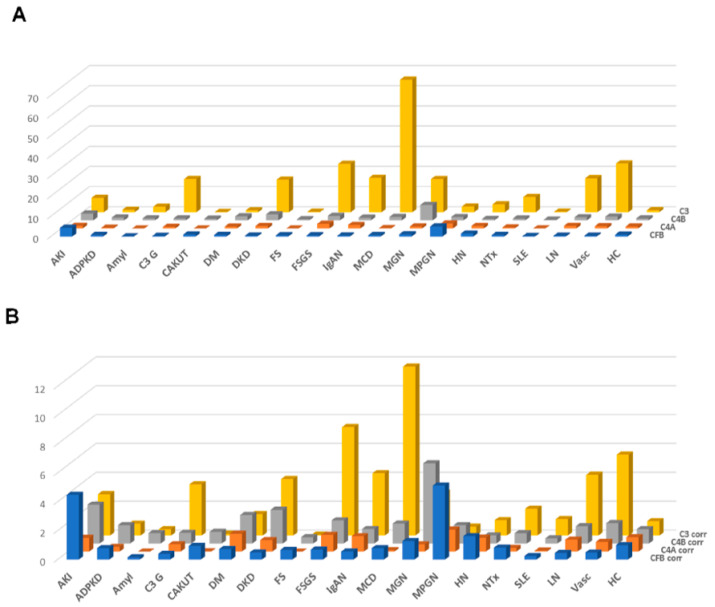
Combined abundances (normalized to the average abundance in healthy controls) of the complement fragments from the four different members of the complement family, unadjusted or adjusted for proteinuria. (**A**) Combined complement factor peptide abundances, unadjusted. (**B**) Combined complement factor peptide abundances, adjusted for proteinuria. While adjustment for proteinuria induces some changes, the overall distribution is not substantially affected. Abbreviations: ADPKD: Autosomal dominant polycystic kidney disease; AKI: Acute Kidney Injury; Amyl: Amyloidosis; C3G: C3 glomerulopathy; CAKUT: Congenital anomalies of the kidney and urinary tract; DKD: Diabetic kidney disease; DM: Diabetes Mellitus; FS: Fanconi syndrome; FSGS: focal segmental glomerulosclerosis; HC: Healthy control; HN: Hypertensive nephrosclerosis; IgAN: IgA nephropathy; LN: lupus nephritis; MCD: minimal change disease; MGN: membranous glomerulonephritis; MPGN: membranoproliferative glomerulonephritis; NTx: Kidney transplantation; SLE: Systemic lupus erythematosus; Vas: Vasculitis.

**Figure 4 proteomes-09-00049-f004:**
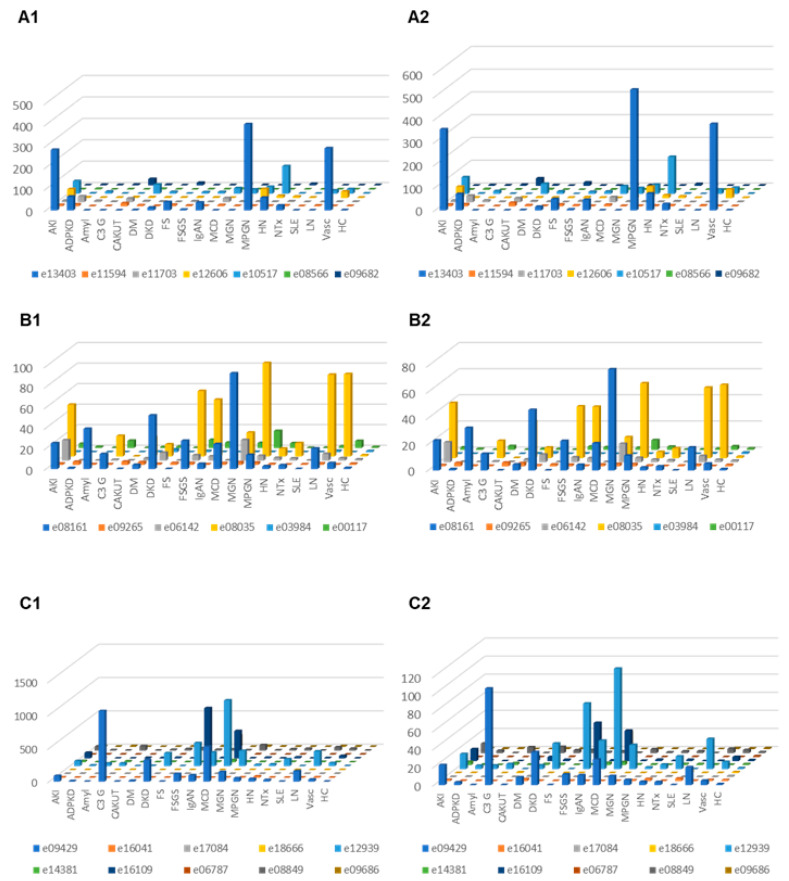
Relative abundances (normalized to the average abundance in healthy controls) of complement fragments from the four different members of the complement family, unadjusted or adjusted for proteinuria. (**A1**) Complement factor B derived peptides, unadjusted. (**A2**) Complement factor B derived urine peptides, adjusted for proteinuria. (**B1**) Peptides from complement factor 4A and 4B, unadjusted. (**B2**) Complement factor 4A and 4B, adjusted for proteinuria. (**C1**) Peptides from complement factor 3, unadjusted. (**C2**) Peptides from complement factor 3, adjusted. Abbreviations: ADPKD: Autosomal dominant polycystic kidney disease; AKI: Acute Kidney Injury; Amyl: Amyloidosis; C3G: C3 glomerulopathy; CAKUT: Congenital anomalies of the kidney and urinary tract; DKD: Diabetic kidney disease; DM: Diabetes Mellitus; FS: Fanconi syndrome; FSGS: focal segmental glomerulosclerosis; HC: Healthy control; HN: Hypertensive nephrosclerosis; IgAN: IgA nephropathy; LN: lupus nephritis; MCD: minimal change disease; MGN: membranous glomerulonephritis; MPGN: membranoproliferative glomerulonephritis; NTx: Kidney transplantation; SLE: Systemic lupus erythematosus; Vas: Vasculitis.

**Table 1 proteomes-09-00049-t001:** List of detected complement fragments.

Peptide ID	Sequence	Complement	Start AA	Stop AA	Avg. rel. abund.	rho eGFR	*p*-Value eGFR	rho PU	*p*-Value PU
e13403	FLSSLTETIEGVDAEDGHGPGEQQ	CFB	234	257	51.86	−0.053	0.5059	0.174	0.0482
e11594	LSSLTETIEGVDAEDGHGPGEQ	CFB	235	256	186.82	0.19	<0.0001	−0.06	0.0276
e11703	SSLTETIEGVDAEDGHGPGEQQ	CFB	236	257	79.35	0.236	<0.0001	−0.133	0.0565
e12606	LSSLTETIEGVDAEDGHGPGEQQ	CFB	236	257	876.10	−0.044	0.0069	0.074	0.0004
e10517	LTETIEGVDAEDGHGPGEQQ	CFB	238	257	59.02	0.298	<0.0001	−0.178	<0.0001
e08566	TETIEGVDAEDGHGPGEQ	CFB	239	256	434.59	0.269	<0.0001	−0.203	<0.0001
e09682	TETIEGVDAEDGHGPGEQQ	CFB	239	257	109.77	0.377	<0.0001	−0.063	0.1778
e08161	TLTKAPADLRGVAHNNL	C4B	1201	1217	51.00	0.056	0.2757	−0.012	0.8334
e06142	TKAPADLRGVAHNNL	C4B	1203	1217	98.92	−0.175	<0.0001	0.224	<0.0001
e09265	DELPAKDDPDAPLQPVTP	C4B	1423	1440	134.67	0.299	<0.0001	−0.261	<0.0001
e08035	TLTKAPVDLLGVAHNNL	C4A	1201	1217	74.57	0.123	0.0422	−0.08	0.2586
e03984	APVDLLGVAHNNL	C4A	1205	1217	22.80	0.063	0.5361	0.051	0.6521
e00117	LGVAHNNL	C4A	1210	1217	116.12	−0.041	0.1899	0.326	<0.0001
e09429	EGVQKEDIPPADLSDQVP	C3	955	972	882.83	−0.156	0.0001	0.254	<0.0001
e16041	EGVQKEDIPPADLSDQVPDTESETRIL	C3	955	981	169.49	0.059	0.0028	0.038	0.1365
e17084	EGVQKEDIPPADLSDQVPDTESETRILLQ	C3	955	983	56.92	0.356	<0.0001	−0.176	<0.0001
e18666	EGVQKEDIPPADLSDQVPDTESETRILLQGTPVA	C3	955	988	4.08	0.126	0.1361	−0.17	0.1261
e12939	LQGTPVAQMTEDAVDAERLKHL	C3	982	1003	3066.93	−0.213	<0.0001	0.445	<0.0001
e06787	IGGLRNNNEKDMALT	C3	1130	1144	154.90	−0.005	0.8695	0.103	0.0147
e14381	LTTAKDKNRWEDPGKQLYNVEAT	C3	1211	1233	79.81	−0.28	<0.0001	0.278	<0.0001
e16109	LTTAKDKNRWEDPGKQLYNVEATSYA	C3	1211	1236	74.69	−0.148	0.0264	0.148	0.0524
e09686	QALAQYQKDAPDHQELN	C3	1277	1293	152.22	−0.23	<0.0001	0.359	<0.0001
e08849	YQKDAPDHQELNLDVS	C3	1282	1297	80.10	−0.021	0.7393	0.082	0.2554

Given are an identification number (ID), amino acid sequence, parental complement protein, peptide position in the protein sequence, and average relative abundance of these peptides calculated based on the full dataset of 16,027 individuals, along with correlation coefficient (rho) with eGFR and proteinuria (PU) of complement fragments together with the respective *p*-value. Abbreviations: AA: amino acid; Avg. rel. abund.: Average relative abundance; C3: Complement C3; C4B: Complement 4B; CFB: Complement Factor B; eGFR: estimated glomerular filtration rate; PU: proteinuria; rho: Spearman’s Rank Correlation Coefficient.

**Table 2 proteomes-09-00049-t002:** Distribution conditions/etiologies for urinary peptidomics datasets included in the study.

Disease/Condition	N
Acute kidney injury	422
ADPKD	273
Amyloidosis	8
Atypical hemolytic uremic syndrome	8
C3 glomerulopathy	24
CAKUT	567
Diabetes Mellitus	4428
Diabetic kidney disease	1401
Fanconi syndrom	12
FSGS	126
IgAN	811
MCD	50
MGN	113
Morbus Fabry	66
MPGN	50
Hypertensive nephrosclerosis	104
Nephrotic syndrom	127
Kidney transplantation	2300
SLE	20
LN	94
Vasculitis	159
Healthy control	4864
Total	16,027

Abbreviations: ADPKD: Autosomal dominant polycystic kidney disease; CAKUT: Congenital anomalies of the kidney and urinary tract; FSGS: focal segmental glomerulosclerosis; IgAN: IgA nephropathy; LN: lupus nephritis; MCD: minimal change disease; MGN: membranous glomerulonephritis; MPGN: membranoproliferative glomerulonephritis; SLE: Systemic lupus erythematosus.

## Data Availability

Anonymised data will be made available upon request directed to the corresponding author. Proposals will be reviewed and approved by the investigators and collaborators based on scientific merit. After approval of a proposal, data will be shared through a secure online platform after signing a data access and confidentiality agreement. Mass spectrometry data (LC-MS/MS) of representative samples used for sequencing of complement peptides are deposited at Zenodo (doi:10.5281/zenodo.5713591).

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
