# Peer review of "Molecular Mapping of Urinary Complement Peptides in Kidney Diseases"

_proteomes, 2021, doi:10.3390/proteomes9040049_

Round 1

Reviewer 1 Report

The proteins identified were not confirmed by other methods, such as WB or ELISA.

As a result, none of the protein expression is ensured.

The authors should perform more experiments.

Author Response

Comment 1: The proteins identified were not confirmed by other methods, such as WB or ELISA. As a result, none of the protein expression is ensured. The authors should perform more experiments.

Response to Comment 1: The manuscript reports on naturally occurring peptides derived from complement proteins. Unfortunately, no antibodies are available targeting specifically these peptides, hence, it is not possible to perform any ELISA assays. Western blots are generally not applicable for peptide analysis. We are not addressing expression, but presence of peptides, which is the result of protease activity (although, of course, in order to be cleaved, the protein has to be present, which initially requires expression).

Reviewer 2 Report

The authors did peptidomics analysis of urine in different human kidny diseases and hypothesized multiple complement-derived peptides were significantly correlated to kidney function and aetiologies.

 (1) Line 137-138: Since there is only one line for “Comple-ment” in Table 1, remove the “-”. And the notes below the table mentioned “peptide mass [Da], migration[min] ”, no information of these two items are in Table 1, please add these information for each peptide.

(2) Line 167: Since all the of “association” in A,B,D figures are not capital, change the “Association” in figure C to “association” for consistency.

(3) Line 181-182: According to the description, the order is CFB-C4A-C4B-C3, which is not consist with figure order, please switch C4A and C4B in either way.

(4) No description for Figure 3B in the main text.

(5) A1 and A2 labeling were lost in Figure 4, please add them into the figure.

(7) Line 382: There are two periods in the end, remove one of them.

(6) Please upload raw files to ProteomeXchange and add the accession number into the paper.

(7) A suggestion for future work: Protein level change, validation of the peptides/proteins with kidney functions/diseases.

Author Response

General Comment: The authors did peptidomics analysis of urine in different human kidny diseases and hypothesized multiple complement-derived peptides were significantly correlated to kidney function and aetiologies.

Comment 1: Line 137-138: Since there is only one line for “Comple-ment” in Table 1, remove the “-”. And the notes below the table mentioned “peptide mass [Da], migration[min] ”, no information of these two items are in Table 1, please add these information for each peptide.

Response to Comment 1: Thank you very much for spotting this mistake, which now has been corrected. We deleted the reference to mass and migration time in CE since this information does not appear of any value in the context of this study.

Comment 2: Line 167: Since all the of “association” in A,B,D figures are not capital, change the “Association” in figure C to “association” for consistency.

Response to Comment 2: We apologize for this mistake, which now has been corrected.

Comment 3: Line 181-182: According to the description, the order is CFB-C4A-C4B-C3, which is not consist with figure order, please switch C4A and C4B in either way.

Response to Comment 3: We apologize for this mistake, which now has been corrected in the legend

Comment 4: No description for Figure 3B in the main text.

Response to Comment 4: Thank you for the comment. We apologize for this error and have corrected it in the revised manuscript.

Comment 5: A1 and A2 labeling were lost in Figure 4, please add them into the figure.

Response to Comment 5: We apologize for this mistake, which now has been corrected.

Comment 6: Line 382: There are two periods in the end, remove one of them.

Response to Comment 6: Thank you for the careful evaluation of our manuscript. We are sorry for the mistake, it is now corrected.

Comment 7: Please upload raw files to ProteomeXchange and add the accession number into the paper.

Response to Comment 7: We have uploaded the data on Zenodo (https://zenodo.org/) and now give the ID of the uploaded data in the paper (doi: 10.5281/zenodo.5713591).

Comment 8: A suggestion for future work: Protein level change, validation of the peptides/proteins with kidney functions/diseases.

Response to Comment 8: We thank the Reviewer for these helpful suggestions and hope to report on the results in a separate manuscript.

Reviewer 3 Report

Comment to the Author

  1. The author should describe more detail at Material and Methods. Especially, number of samples, MS setting, how analyzed data .
  2. The writing often lacks clarity and sharpness, and several sections need to be improved (e.g., Line382 “approach..”. Furthermore, the author should fix some of the sentences in manuscript for improving English.
  3. Fig2 is difficult to see cause of hiding by bars. If it possible, prepare by another graph.
  4. The author should indicate the correlation coefficient at Figure 1. Then, discuss.

Comment to the Author

  1. The author should describe more detail at Material and Methods. Especially, number of samples, MS setting, how analyzed data .
  2. The writing often lacks clarity and sharpness, and several sections need to be improved (e.g., Line382 “approach..”. Furthermore, the author should fix some of the sentences in manuscript for improving English.
  3. Fig2 is difficult to see cause of hiding by bars. If it possible, prepare by another graph.
  4. The author should indicate the correlation coefficient at Figure 1. Then, discuss.

Author Response

Comment 1: The author should describe more detail at Material and Methods. Especially, number of samples, MS setting, how analyzed data.

Response to Comment 1: We thank the Reviewer for this comment and have now substantially expanded the section on MS analysis in the methods.

Comment 2: The writing often lacks clarity and sharpness, and several sections need to be improved (e.g., Line382 “approach..”. Furthermore, the author should fix some of the sentences in manuscript for improving English.

Response to Comment 2: We apologize for the lack of clarity and have tried to revise the manuscript accordingly.

Comment 3: Fig2 is difficult to see cause of hiding by bars. If it possible, prepare by another graph.

Response to Comment 3: This is a well-taken point. We have tried to improve on Figure 2, but the results were not better. It appears not possible to display all columns, owed to the complexity. However, we now specifically refer to this problem in the main text. In addition, to enable detailed inspection of the data, we emphasize in the manuscript that all underlying data are also presented in Supplementary Table 1.

Comment 4: The author should indicate the correlation coefficient at Figure 1. Then, discuss.

Response to Comment 4: Thank you for the comment. We have added the correlation coefficient to all panels in Figure1. We were uncertain what to discuss, beyond the discussion already present in the manuscript. However, we added that in general association with eGFR and proteinuria were found of opposite orientation, as expected, since eGFR is inversely associated with proteinuria.

Reviewer 4 Report

The authors analyzed a large dataset collected by the Human Urine Proteome project and attempted to elucidate the impact of complement system in the context of kidney diseases. The authors provided conclusions, which are supported by the previously reported independent data regarding the role of C3 and CFB, especially, amongst different nephropathies. However, while reading the paper, numerous questions raised regarding data handling and statistical processing, which are key points in this meta-analysis but not completely clear and somewhat doubtful.

  1. First of all, it is not clear the statistical processing of data obtained. How did the authors calculate the combined abundance? If there are several mutual peptides derived from the same region of the certain complement factor (as indicated in Table 1), the abundances of such peptides are summarized. If there are completely different peptides originating from the same complement factor, did you summarize them again into one variable? What is exactly summarized: absolute abundance, normalized relative abundance? If data obtained by different mass spectrometers (on Q-TOFs of different vendors, on an Orbitraps, and both types of MS machine are distinguished by a background intensity and limits of the signal intensity), how did authors handle such data with different absolute intensities?
  2. Why did the authors use such a strange approach to estimate the influence of proteinuria? As far as is stated, the authors have information regarding the creatinine concentration and protein concentration otherwise they could not calculate eGFR. So, why did not employ albumin-to-creatinine or protein-to-creatinine ratio as a corrective measure for consideration? Maybe, that is a more accurate approach to achieve plausible input of different complement factors. For me, it is not clear the rationale of such corrective action used by the authors. How the proteinuria was measured by cofounders: was it one-time samples collected in the Human Urinary Proteome dataset or was it daily urine output?
  3. If patients are characterized by a different number of peptides derived from the same complement factors (for example, three different mutual peptides were observed in patient-A and five mutual peptides were observed in patient-B), does it mean different complements system activity? Or different activity of endogenous proteases? Or insufficient curation of samples collection and storage?
  4. Although the authors suggested, the correlation between CFB and proteinuria is poor (unlike for C3) due to its lower concentration (200 mg/L vice 1000 mg/L), this suggestion seems unplausible because of the difference between C3 and CFB concentration is not significant. I suggest a more mechanistic explanation based on the activities of CFB in the complement system escalation and the related inflammation process. Moreover, the majority of observed complement factors (CFB, C4A, and C4B) demonstrated weak correlation and the main focus of the discussion environment is around the C3 factor. At least the authors have to suggest why did they observe an ambiguous association between different complement factors and eGFR.

Author Response

General Comment: The authors analyzed a large dataset collected by the Human Urine Proteome project and attempted to elucidate the impact of complement system in the context of kidney diseases. The authors provided conclusions, which are supported by the previously reported independent data regarding the role of C3 and CFB, especially, amongst different nephropathies. However, while reading the paper, numerous questions raised regarding data handling and statistical processing, which are key points in this meta-analysis but not completely clear and somewhat doubtful.

Comment 1: First of all, it is not clear the statistical processing of data obtained. How did the authors calculate the combined abundance? If there are several mutual peptides derived from the same region of the certain complement factor (as indicated in Table 1), the abundances of such peptides are summarized. If there are completely different peptides originating from the same complement factor, did you summarize them again into one variable? What is exactly summarized: absolute abundance, normalized relative abundance? If data obtained by different mass spectrometers (on Q-TOFs of different vendors, on an Orbitraps, and both types of MS machine are distinguished by a background intensity and limits of the signal intensity), how did authors handle such data with different absolute intensities?

Response to Comment 1: We thank the Reviewer for these comments. The method section includes all statistical approaches used within this study. Specifically, we have investigated association of complement peptides with clinical data (e.g. eGFR) using Spearman's rank correlation. Regarding the “combined abundance”: We have combined the data from all peptides belonging to the respective complement, as now indicated more clearly in the methods section. Combination was performed by summarizing the relative peptide abundance. We also now indicated how the relative abundance was calculated. We apologize for the unclear description in the manuscript and hope that in the revised manuscript the above mentioned issues are now more clearly stated. Also based on the comments from Reviewer 2, we have added a more detailed description of the acquisition of the MS data. We hope that, based on this detailed description, it is evident in the revised manuscript that only data from one specific type of instrument were used (either Bruker micrOTOF for CE-MS or Thermo Orbitrap for LC-MS/MS to obtain information about peptide sequence).

Comment 2: Why did the authors use such a strange approach to estimate the influence of proteinuria? As far as is stated, the authors have information regarding the creatinine concentration and protein concentration otherwise they could not calculate eGFR. So, why did not employ albumin-to-creatinine or protein-to-creatinine ratio as a corrective measure for consideration? Maybe, that is a more accurate approach to achieve plausible input of different complement factors. For me, it is not clear the rationale of such corrective action used by the authors. How the proteinuria was measured by cofounders: was it one-time samples collected in the Human Urinary Proteome dataset or was it daily urine output?

Response to Comment 2: We apologize for not being clear enough. We hypothesize that complement peptides in urine are in part the result of glomerular filtration, and in part the result of proteinuria, and both of these processes contribute to the final abundance of urinary peptides. Thus, to correct for proteinuria (g/g creatinine) (specifically the fraction of peptides being a result of the proteinuria), we need to first calculate the impact of proteinuria on the complement abundance, followed by its subtraction from the total peptide abundance. This was done using the method presented in the paper. When applying simple correction as suggested by Reviewer, we would assume that abundance of urinary peptides is only attributed to proteinuria, which is not the case. We have tried to clarify the rationales underlying the approach used for the correction for the proteinuria. Proteinuria measurements were the results of turbidimetric assay.

Comment 3: If patients are characterized by a different number of peptides derived from the same complement factors (for example, three different mutual peptides were observed in patient-A and five mutual peptides were observed in patient-B), does it mean different complements system activity? Or different activity of endogenous proteases? Or insufficient curation of samples collection and storage?

Response to Comment 3: This is an interesting question, but unfortunately we have no definitive answer. We do not know the specific processes that result in the generation of the different peptides. The most plausible explanation may be differences in the activity of endogenous proteases that generate the peptides. It also seems plausible that these proteases are not tightly linked to complement activity, but belong to a more generic protein degradation machinery. Based on these thoughts, we assume that the combined relative abundance of the peptides may in fact be well suited to display (previous) complement activation.

Comment 4: Although the authors suggested, the correlation between CFB and proteinuria is poor (unlike for C3) due to its lower concentration (200 mg/L vice 1000 mg/L), this suggestion seems unplausible because of the difference between C3 and CFB concentration is not significant. I suggest a more mechanistic explanation based on the activities of CFB in the complement system escalation and the related inflammation process. Moreover, the majority of observed complement factors (CFB, C4A, and C4B) demonstrated weak correlation and the main focus of the discussion environment is around the C3 factor. At least the authors have to suggest why did they observe an ambiguous association between different complement factors and eGFR.

Response to Comment 4: We respectfully do not agree with the Reviewer that our explanation is unplausible. While there are other explanations as well, we feel the explanation presented makes sense. However, we agree that an explanation based on the activities of CFB in the complement system escalation and the related inflammation is at least equally plausible, and we thank the Reviewer for this suggestion. It seemed best to present both possible explanations in the revised manuscript. We hope that the revision of the manuscript on this topic presents acceptable and plausible hypotheses for the ambiguous association of complement factors with eGFR.

Round 2

Reviewer 1 Report

Although there are no antibodies specifically for these peptides, the authors can still perform ELISA or Western for the intact proteins. This is the minimum requirement to confirm whether there is any change in the protein expression.

Author Response

In general urinary peptides do not reflect urinary protein content, among others a result of peptides passing the filtration barrier, while proteins are retained. Therefore, assessing complement proteins via ELISA cannot support the findings on complement peptides. We also want to point out that in general the selectivity/specificity of the MS technology is superior to antibody-based approaches, especially when investigating peptides.

Reviewer 3 Report

Accept in present form

Author Response

We thank the reviewer for the positive comment.

Reviewer 4 Report

The authors addressed most of concerns and made the text more streamlined for the better understanding. They added essential information and correction in the statistical analysis and brief discussion regarding the possible reasons of peptides pattern heterogeneity. At the present state, after revision, the conclusion looks more powerful and supports the objection between peptides complement-specific patterns and kidney diseases.

However, for the future assay, I would recommend to authors to use some kind of spectra validating source, like Peptide Atlas, for example, which is a part of HUPO projects. While achieving the Supplementary Materials submitted by the authors to Zenodo (DOI: 10.5281/zenodo.5713591), I found a Files Description, listing several complement-related peptides. If checking such peptides through the Peptides Atlas source, you may observe, that some of them are ambiguous or redundant. For example, peptide LTTAKDKNRWEDPGKQLYNVEAT (ID e16109), peptide e14381, peptide e11594, and some other peptides one may find that these peptides belong to MHC I or MHC II complexes, MHC class I HLA-Cw*0803, Small integral membrane protein 14, etc. While keeping in mind amino acids redundancy especially among specific protein classes (like MHC), caution should be paid to possible interference. Even single amino acid residue might be decision-making, like in the case of peptide LSSLTETIEGVDAEDGHGPGEQQ, which is specific for CFB, and peptide LSSLTETIEGVDAEDGHGPGEQ (one C-terminal Q omitted), which is redundant. Any researcher has a flexible choice of how to validate the proposition, but as many possibilities are used as much you enhance the soundness of investigation. A better data curation will provide a great opportunity for the final assumption.

Notwithstanding, the revised paper looks much better for readers and now covers essential explanations in statistics and in physiology and heterogeneity of complement-derived peptides origination regardless of the severity and type of kidney diseases.

Author Response

We thank the reviwer for the positive comments.